# Improving Language Plasticity via Pretraining with Active Forgetting

**Yihong Chen**$^{\Upsilon\delta}$    **Kelly Marchisio**$^{\aleph}$    **Roberta Raileanu**$^{\delta}$    **David Ifeoluwa Adelani**$^{\Upsilon}$
**Pontus Stenetorp**$^{\Upsilon}$    **Sebastian Riedel**$^{\Upsilon}$    **Mikel Artetxe**$^{\aleph\aleph}$

$^{\Upsilon}$UCL Centre for Artificial Intelligence
$^{\delta}$Meta AI    $^{\aleph\aleph}$Reka AI    $^{\aleph}$Cohere AI

{yihong.chen, d.adelani, p.stenetorp, s.riedel}@cs.ucl.ac.uk
mikel@reka.ai    kelly@cohere.com    raileanu@meta.com

## Abstract

Pretrained language models (PLMs) are today the primary model for natural language processing. Despite their impressive downstream performance, it can be difficult to apply PLMs to new languages, a barrier to making their capabilities universally accessible. While prior work has shown it possible to address this issue by learning a new embedding layer for the new language, doing so is both data and compute inefficient. We propose to use an *active forgetting* mechanism during pretraining, as a simple way of creating PLMs that can quickly adapt to new languages. Concretely, by resetting the embedding layer every $K$ updates during pretraining, we encourage the PLM to improve its ability of learning new embeddings within limited number of updates, similar to a meta-learning effect. Experiments with RoBERTa show that models pretrained with our forgetting mechanism not only demonstrate faster convergence during language adaptation, but also outperform standard ones in a low-data regime, particularly for languages that are distant from English. Code will be available at https://github.com/facebookresearch/language-model-plasticity.

## 1 Introduction

Pretrained language models (PLMs) have been swiftly reshaping the landscape of natural language processing (NLP) by improving upon standardized benchmarks across the board [Radford and Narasimhan, 2018, Devlin et al., 2019, Liu et al., 2019, Brown et al., 2020]. At their core, they acquire knowledge by ingesting large datasets and store this knowledge in their parameters during pretraining. Using finetuning or prompting [Brown et al., 2020], such knowledge can then be applied to downstream applications, such as semantic analysis, question answering, and others.

Despite their success, PLMs still have a number of shortcomings [Weidinger et al., 2021, 2022]. In particular, it requires massive data and computation to pretrain them [Gururangan et al., 2020, Kaplan et al., 2020, Hernandez et al., 2021, Hu et al., 2021, Touvron et al., 2023]. Naively retraining a new PLM to accommodate every lingual space shift [1] would be prohibitively expensive. This makes it a highly relevant research target to create PLMs that can be efficiently adapted to new lingual spaces.

While forgetting in the context of both human and machine learning is often perceived as something negative (for instance catastrophic forgetting [McCloskey and Cohen, 1989, Ratcliff, 1990, Kirkpatrick et al., 2017]), recent works have shown that for artificial neural networks forgetting

---

[1]We use the term *lingual space shift* to describe changes in language usage between pretraining and the target downstream application, caused by factors such as language change, time evolution, or domain switch. A model with high *language plasticity* would quickly adapt to these shifts.

can also play a positive role in increasing their "plasticity", such as improving generalization to unseen data [Zhou et al., 2022, Chen et al., 2022, Igl et al., 2021], enabling learning in low-data regimes [Alabdulmohsin et al., 2021, Taha et al., 2021], or counteracting primacy bias [Nikishin et al., 2022, D'Oro et al., 2023]. Given these developments, in this work we ask whether we can draw upon forgetting as a mechanism to improve *pretraining* and imbue PLMs with similar benefits.

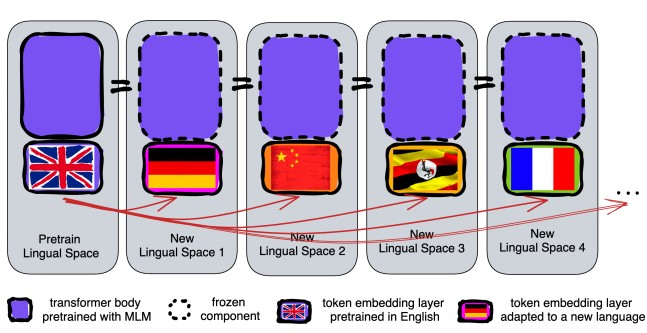

Figure 1: *Rewiring* via relearning token embeddings: where the transformer body (the purple part) is "frozen" and reused for a new language, but the token embeddings are relearned to suit the new language.

It is well established in the NLP community that models struggle to generalize across languages without substantial intervention [Conneau et al., 2020, Pfeiffer et al., 2020, 2022, Ansell et al., 2022], which is especially true for low-resources languages. We thus see this as a promising testing ground for forgetting techniques. Our focus is on the input layer of the PLM, the *token embedding layer*, as learning it has been shown to be highly effective when adapting between languages [Artetxe et al., 2020].

Concretely, we introduce an *active forgetting* mechanism, that resets token embeddings at regular intervals, while leaving all other parameters untouched throughout pretraining. We study whether this forgetting approach creates a PLM that can easily *rewire* (Figure 1) to an unseen (possibly distant) language. Intuitively, resetting embeddings forces the transformer body to re-derive reasoning each time instead of relying on memorized shortcuts. Through repetition, the body learns more abstract, high-level reasoning. A model with greater abstraction can easily transfer across languages, since high-level reasoning is more language-agnostic.

Our zero-shot evaluations on several cross-lingual transfer benchmarks show that for cases where unlabeled adaptation corpus for the unseen language has as few as 5 million tokens (a low-data regime), forgetting PLMs outperforms the baseline by large margins: average gains of +21.2% on XNLI, +33.8% on MLQA, and +60.9% on XQuAD. In addition, models pretrained using active forgetting converge faster during language adaptation. Finally, we find that forgetting is especially beneficial for languages that *are distant from* English, such as Arabic, Hindi, Thai, and Turkish.

## 2 Rewire PLMs for New Languages

Using unlabeled data, Artetxe et al. [2020] demonstrates the possibility of rewiring a monolingual PLM for a new language; they propose to relearn the embedding layer for the new language while keeping all the other parameters frozen. The underlying assumption is that the token embedding layer and the transformer body (the non-token-embedding parameters) divide up the responsibility in a way that the former handles language-specific lexical meanings, while the latter deals with high-level general reasoning. Hence, rewiring an English PLM for a new language boils down to separately adapting the former with unlabeled data in the new language and the latter with English task data. The procedure can be summarized as follows:

1. Pretrain: A transformer-based model is pretrained on an *English* corpus. In our experiments, we choose to pretrain RoBERTa-base Liu et al. [2019], a 12-layer transformer-based model, on English CC100 [Conneau et al., 2020].

2. Language Adapt: The token embedding layer is finetuned using unlabelled data in the new language, while the transformer body is frozen.

3. Task Adapt: The transformer body is finetuned using downstream task data in English, while the token embedding layer is frozen.

4. Assemble: The final model is assembled by taking the adapted token embedding layer from stage 2 and the adapted transformer body from stage 3.

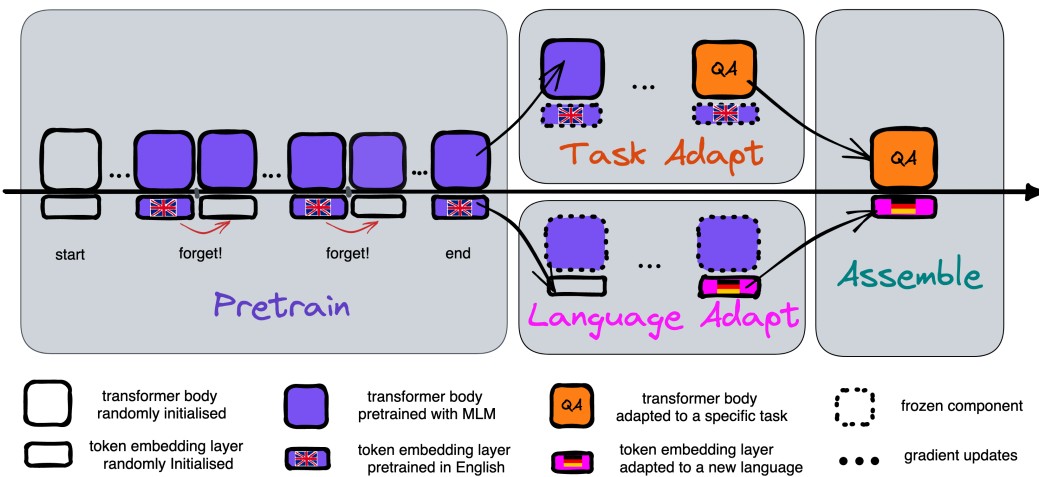

Figure 2: Unsupervised zero-shot cross-lingual transfer. **Left**: in the pretrain stage, we compare standard pretraining with forgetting pretraining, where the token embeddings are actively forgotten at a regular interval while the transformer body is learned as the standard pretraining. **Middle**: the task adapt stage and the language adapt stage separately adapt the transformer body using English task data and the token embeddings using unlabeled data in the new language. **Right**: the assemble stage reassemble the adapted body and token embedding layer into a usable PLM.

## 2.1 On The Difficulty of Rewiring PLMs via Relearning the Token Embeddings

While the above procedure [Artetxe et al., 2020] offers a general framework for rewiring a monolingual PLM with unlabelled data in the new language, it is unclear how efficient such rewiring can be, including both sample efficiency and computational efficiency. To better understand the difficulty of rewiring PLMs via relearning the token embeddings, we design an experiment where we relearn the token embedding layer using varying amounts of adaptation data. For illustration purpose, we pick English as the pseudo "adaptation language" because the English dataset is large enough to bootstrap a series of sub-datasets with varying quantity. We create sub-datasets with $[1K, 10K, 100K, 1M, 5M, 10M, 100M, 1B, 10B]$ tokens and relearn the English embeddings while keeping the transformer body frozen.

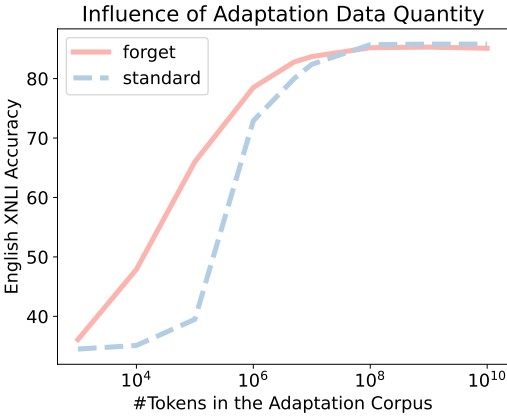

Figure 3: The rewiring performance for standard PLMs (blue dashed line) drops drastically if the adaptation tokens $\leq$ 10M.

The dashed blue line in Figure 3 summarizes the influence of the adaptation data quantity on the quality of the rewired PLMs (relearned embeddings assembled with the English NLI task body). We can see that the standard PLMs are easy to rewire if there is enough adaptation data. However if the adaptation corpus contains fewer than 10 million tokens, the performance of the rewired standard PLMs (the blue dashed line in the figure) drops drastically as the adaptation data quantity goes down, from near 80 to around 35, a random-guessing level for NLI tasks. This motivates us to create more rewirable PLMs, i.e. PLMs with more plasticity so that the rewiring process can be faster and consume less data.

# 3 Pretrain Easy-to-Rewire PLMs via Active Forgetting

Recent works have shown that incorporating forgetting through iterative weights resetting can increase the "plasticity" of neural networks, enabling them to learn from small data and generalize better to unseen data in supervised learning [Alabdulmohsin et al., 2021, Taha et al., 2021, Zhou et al., 2022]. Building on these efforts, we study if we can bring such forgetting into the pretrain stage so that the resulting PLM would have more plasticity, allowing easier rewiring to new languages.

**Our Hypothesis.** In effect, when Artetxe et al. [2020] relearned the token embedding layer, the reinitialization of the embeddings can be seen as forgetting applied *once* at the start of the language adapt stage. However, the PLM (specifically the transformer body) has never encountered forgetting before this stage and may struggle to handle this new situation. Without early exposure to forgetting, the PLM might suffer from slow recovery caused by forgetting before eventually benefiting from it. The learning of a new lexical embedding layer in a PLM henceforth consumes lots of data in new languages along with long training horizons as shown in Section 2.1. In this paper, to ensure swift learning of the new languages with both high sample efficiency and convergence rate, we argue that the PLM must be exposed to forgetting during pretraining, allowing itself to maximize the positive impact of forgetting and minimizing the cost of recovery.

**Our Method.** With this hypothesis in mind, we propose to add an *active forgetting* mechanism to the pretraining procedure, which resets the token embedding layer periodically as described in Algorithm 1. Concretely, the forgetting mechanism operates by intentionally clearing the weights of the embedding layer, which stores the static representations for all tokens, and reinitializing them to a new set of random values every $K$ gradient updates. Since pretraining involves advanced training strategies, like optimizers with states and learning rate schedulers, we also reset them together with the token embedding layer. We refer to language models pretrained with such active forgetting mechanism as *forgetting PLMs*, in contrast to *standard PLMs* which are pretrained in a standard way. The pretraining loss curve of a forgetting PLM is episodic (Figure 4), like in reinforcement learning or meta-learning. This episodic learning demonstrates that the active forgetting mechanism can introduce diversity without requiring actual new data. Each forgetting event kind of "branches out" a novel environment for the model to explore, as if initiating a new episode of learning.

**Research Questions.** To further examine the proposed forgetting mechanism, we compare *forgetting PLMs* and *standard PLMs* on sample efficiency and convergence speed during language adapt, two key aspects of model plasticity. Our research investigates:

- RQ1: Real-world low-resource languages often have scarce data for adapting models. Does pretraining with active forgetting impart enough plasticity to forgetting PLMs, enabling them to learn new languages even with such limited data?

- RQ2: Deploying PLMs frequently encounters computational limitations. Endowed with more plasticity, can forgetting PLMs reduce adaptation time for such low-compute scenarios?

- RQ3: New languages may be very similar or different from pretraining languages. Does this similarity/difference impact the relative benefit of forgetting PLMs over standard PLMs?

# 4 Evaluate Forgetting PLMs for Unsupervised Cross-Lingual Transfer

To evaluate the effectiveness of forgetting PLMs and address RQ1-RQ3, we conduct experiments on several cross-lingual transfer benchmarks.

## 4.1 Experimental Setup

In our work, we closely follow the setup in Artetxe et al. [2020] and Marchisio et al. [2022]. Our pretraining model is RoBERTa-base, a standard 12-layer transformer-based language model. We trained language-specific sentencepiece tokenizers [Kudo and Richardson, 2018] with a vocabulary size of 50K over the corresponding data subsets in CC100. The model was pretrained with the English subset of the CC-100 dataset. The pretraining process consists of 125K updates, with a batch size of 2048. We used a learning rate scheduler with linear decay and an initial learning rate of $7e - 4$, with

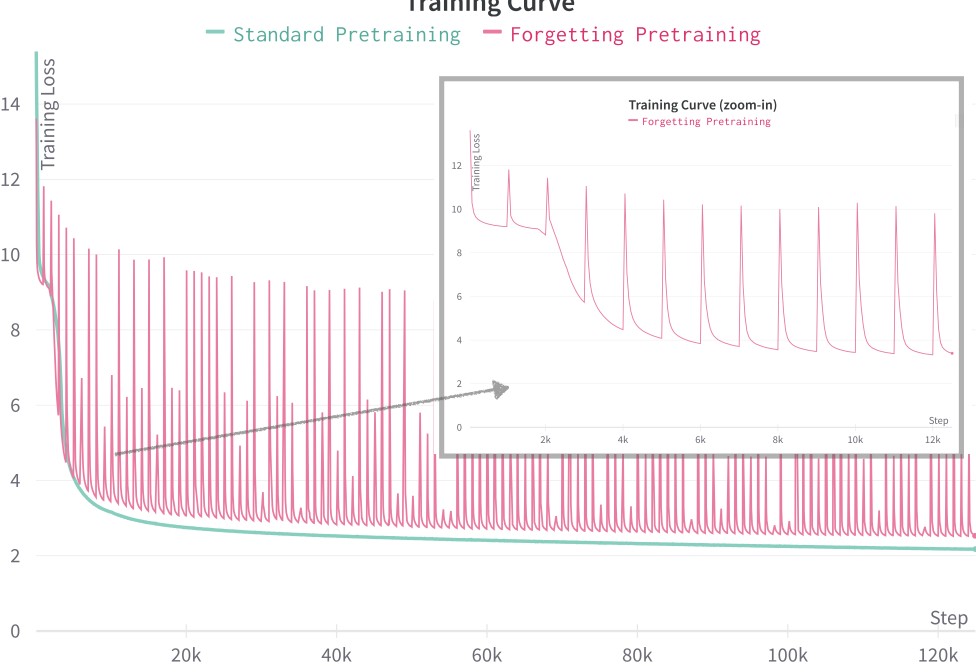

Figure 4: Pretraining loss curves of forgetting and standard language models. The forgetting mechanism brings an episodic pattern into the loss curve: every embedding forgetting produces a loss spike, from which the model learn to recover. Through such repeats of forget-relearn, the model gets used to learn new embeddings from scratch.

---

**Algorithm 1** Active forgetting mechanism. The token embedding layer is reset every $K$ updates.

---

**Input:** $K$, interval between two consecutive forgetting; $n_{\text{body/emb}}$, current effective number of updates for the body or the token embedding layer; $\alpha_{\text{body/emb}}$, current learning rate for the body or the token embedding layer; $P^n_{\text{body/emb}}$, parameters after the $n^{\text{th}}$ update for the body or the token embedding layer; $O^n_{\text{body/emb}}$, optimizer states after the $n^{\text{th}}$ update for the body or the token embedding layer; $\Theta$, randomly initialized embedding parameters, each element drawn from $\mathcal{N}(0, 0.02)$; $f$, the function that computes the gradients w.r.t the parameters using the sampled data; $g$, the function that updates the parameters based on the gradients (e.g. one step in Adam optimizer) $s$, the function that updates the learning rate (e.g. one step in the polynomial learning rate scheduler).

**Output:** the updated parameters and optimizer states $P^{(n)} = \{P^{(n)}_{\text{emb}}, P^{(n)}_{\text{body}}\}$, $O^{(n)} = \{O^{(n)}_{\text{emb}}, O^{(n)}_{\text{body}}\}$.

1: $n_{\text{emb}} = n \bmod K$
2: $\alpha_{\text{body}} = s(n_{\text{body}})$ {adjust learning rate based on $n$}
3: $\alpha_{\text{emb}} = s(n_{\text{emb}})$
4: $G^{(n)} = f(P^{(n-1)}, \cdot)$ {compute all gradients}
5: $P^{(n)}_{\text{body}}, o^{(n)}_{\text{body}} = g(G^{(n)}_{\text{body}}, P^{(n-1)}_{\text{body}}, o^{(n-1)}_{\text{body}}, \alpha_{\text{body}}, n)$ {update the transformer body}
6: **if** $n_{\text{emb}} == 0$ **then**
7: $\quad P^{(n)}_{\text{emb}} = \Theta$ {periodical reset token embeddings and the relevant optimizer states}
8: $\quad o^{(n-1)}_{\text{emb}} = 0$
9: **end if**
10: $P^{(n)}_{\text{emb}}, o^{(n)}_{\text{emb}} = g(G^{(n)}_{\text{emb}}, P^{(n-1)}_{\text{emb}}, o^{(n-1)}_{\text{emb}}, \alpha_{\text{emb}}, n_{\text{emb}})$ {update the token embeddings}

---

| Method | vi | sw | es | bg | de | fr | el | ru |
|---|---|---|---|---|---|---|---|---|
| Standard | **65.8** | 55.6 | 68.0 | 65.5 | 62.2 | 63.5 | 63.1 | 56.9 |
| Forgetting | 62.8 | **59.5** | **74.0** | **71.7** | **68.5** | **71.2** | **70.8** | **65.8** |
| Relative Gain | −4.6% | +7.0% | +8.8% | +9.5% | +10.1% | +12.1% | +12.2% | +15.6% |

Table 1: Accuracy comparison of forgetting and standard PLMs on XNLI (table continues).

| Method | zh | ur | hi | tr | ar | th | Avg | en |
|---|---|---|---|---|---|---|---|---|
| Standard | 53.2 | 36.8 | 39.7 | 38.9 | 41.2 | 35.3 | 53.3 | **86.1** |
| Forgetting | **63.5** | **45.8** | **52.9** | **52.7** | **59.5** | **59.7** | **62.7** | 85.1 |
| Relative Gain | +19.4% | +24.5% | +33.2% | +35.5% | +44.4% | +69.1% | +21.2% | −1.2% |

Table 2: Accuracy comparison of forgetting and standard PLMs on XNLI (table continued). On average, forgetting PLMs achieve a $21.2\%$ relative gain in accuracy compared to standard PLMs across the languages tested, where averaged relative gain $= \frac{\sum_{x \in \{\text{languages}\}} \text{Relative Gain of } x}{\#\text{Languages}}$.

10K warm-up updates. Checkpoints were saved every 500 updates and we always choose the last pretraining checkpoint where possible for optimal performance. For forgetting pretraining, we chose the checkpoint corresponding to the best validation perplexity since the last checkpoint might have token embeddings reset. We set the frequency of forgetting K = 1000 and used a clip-norm of 0.5.

During the language adapt stage, we kept most of the hyper-parameters the same as for pretraining. We finetuned the token embedding layer while keeping the others frozen, as described in Section 2. Note that *no* forgetting happens during this stage because we want the models to learn the new languages as well as possible. In the task adapt stage, both models were finetuned for 10 epochs on the English task data, specifically MultiNLI [Williams et al., 2018] for the NLI task and SQUAD [Rajpurkar et al., 2016] for the QA task. After the assemble stage, we evaluate the zero-shot performance of the assembled model on XNLI [Conneau et al., 2018], a cross-lingual NLI task, along with XQuAD [Artetxe et al., 2020] and MLQA [Lewis et al., 2020], two cross-lingual QA tasks. We report the NLI accuracy and QA F1 on the test sets.

Our experiments were implemented using fairseq [Ott et al., 2019]. The pretraining and language adaptation experiments were conducted on 32 Tesla V100 GPUs (each with 32 GB memory) and took approximately 24-36 hours to complete. The time taken for both stages were quite close to each other even though the latter only involved tuning the embeddings. This demonstrates the importance of reducing the computational cost of the language adaptation stage.

Differing from prior work [Artetxe et al., 2020, Marchisio et al., 2022], we focus on language adapt in low-data regimes. We simulate low-resources scenarios by limiting the adaptation data for each downstream language to only 5M subword tokens from CC100. This is in contrast with conventional setups, where all the tokens in the corresponding languages in CC100 are used for language adaptation. As Table 6 shows, such setups consume several orders of magnitude more data than our 5M-token setup; for instance, the Swahili CC100 subset contains 345M tokens, roughly 69 times larger than our corpus, and the Russian subset contains 34.9B tokens, roughly 6, 980 times larger. Therefore, PLMs that can successfully learn new languages with rich data under traditional setups may struggle to do so with our limited 5M-token corpus.

## 4.2 Forgetting PLMs Work Better in Low-Data Regimes (RQ1)

Standard PLMs struggle in low-data language adaptation, dropping from 86.1 English NLI accuracy to just 53.3 average accuracy on XNLI with limited 5M token adaptation data. Compared to prior work which uses full data from Wikipedia [Artetxe et al., 2020] or from CC100 [Marchisio et al., 2022], the average accuracy on XNLI drops about 18% (from 66.8/66.3 to 53.3). This indicates standard PLMs are not coping well with the low-data regime. In contrast, forgetting PLMs achieve decent 62.7 average XNLI accuracy, a +21.2% relative gain over standard PLMs, as shown in Table 2.

Forgetting PLMs also outperform standard PLMs on MLQA and XQuAD, with average F1 relative gains of +33.8% and +60.9% across languages, as respectively demonstrated in Table 3 and Table 4. Across NLI and QA tasks, forgetting PLMs consistently surpass standard PLMs in low-data regimes.

| Method | es | vi | de | zh | hi | ar | Avg | en |
|---|---|---|---|---|---|---|---|---|
| Standard | 49.4 | 38.3 | 45.3 | 34.1 | 17.7 | 20.8 | 34.3 | **78.9** |
| Forgetting | **55.3** | **45.0** | **53.4** | **43.0** | **28.8** | **34.7** | **43.4** | 78.3 |
| Relative Gain | +12.0% | +17.6% | +17.8% | +26.2% | +62.5% | +67.0% | +33.8% | −0.8% |

Table 3: F1-score comparison of forgetting and standard PLMs on MLQA. On average, forgetting PLMs achieve a $33.8\%$ relative gain in F1 compared to standard PLMs across the languages tested, where averaged relative gain $= \frac{\sum_{x \in \{\text{languages}\}} \text{Relative Gain of } x}{\#\text{Languages}}$.

| Method | vi | es | ru | de | el | zh | hi | ar | th | tr | Avg |
|---|---|---|---|---|---|---|---|---|---|---|---|
| Standard | 49.7 | 57.7 | 49.4 | 50.9 | 48.5 | 32.4 | 21.4 | 22.2 | 15.4 | 13.0 | 36.1 |
| Forgetting | **52.9** | **64.6** | **56.5** | **60.9** | **59.9** | **43.7** | **33.3** | **38.7** | **38.4** | **41.4** | **49.0** |
| Relative Gain | +6.4% | +12.0% | +14.5% | +19.7% | +23.6% | +34.6% | +55.8% | +74.2% | +149.7% | +218.8% | +60.9% |

Table 4: F1-score comparison of forgetting and standard PLMs on XQuAD. On average, forgetting PLMs achieve a $60.9\%$ relative gain in F1 compared to standard PLMs across the languages tested, where averaged relative gain $= \frac{\sum_{x \in \{\text{languages}\}} \text{Relative Gain of } x}{\#\text{Languages}}$.

Why do forgetting PLMs handle the low-data regime better? We hypothesize this is because forgetting PLMs are more robust to different embedding initializations. They encode more universal knowledge in the transformer body. Standard PLMs may encode more "shortcut" knowledge relying on certain embedding initializations. In low data, standard PLMs cannot adjust embeddings towards shortcut routes without access to enough data. Forgetting PLMs do not rely on shortcuts so perform better.

### 4.3 Forgetting PLMs Learn New Languages with Fewer Parameter Updates (RQ2)

We are also interested in how quickly forgetting PLMs and standard PLMs can learn new languages. Figure 5 summarizes adaptation curves on XNLI, MLQA and XQuAD, with each point representing the averaged performance across all languages. In just 5K steps ($4\%$ of full adaptation), forgetting PLMs reach 57.8 accuracy on XNLI while standard PLMs struggle at random guessing levels of 37.2. Similar trends hold for MLQA and XQuAD. After 5K steps, forgetting PLMs achieve 92% of their full performance on XQuAD versus just 53% for standard PLMs (see the last plot in Figure 5).

Why do forgetting PLMs converge faster? We hypothesize it is because periodical embedding resetting forces the body to gradually locate itself on a particular manifold, where it can easily cooperate with new embeddings. This makes the body encourage larger embedding updates when adapting to new languages. Active forgetting simulates language switching during pretraining[2] introducing diversity without new data. This allows faster adaptation to real new languages.

### 4.4 Languages That Are Distant To English Benefit Most From Forgetting PLMs (RQ3)

Up to this point, we have primarily presented the averaged performance. In this section, we delve into a detailed comparison of language-specific performance between forgetting PLMs and standard PLMs on XNLI, MLQA, and XQuAD. To gain a deeper insight into which languages benefit the most from the use of forgetting, we present the relative performance changes in Figure 6 for XNLI and MLQA. The results for XQuAD can be found in Figure 8 in the appendix. Across the spectrum of languages (Table 5), we observe that forgetting provides greater benefits for languages distant to the pretraining language (English) in terms of language family, script and morphology. Specifically, forgetting brings large relative gains for languages such as *Arabic*, *Hindi*, *Thai*, *Turkish*, and *Urdu* compared to closer languages like *German*. Script seems important - forgetting helps Vietnamese and Swahili less despite their distance from English, likely due to the shared Latin script. Examining adaptation curves within the first 5K steps, forgetting PLMs reach substantially superior performance over standard PLMs for almost all languages except Urdu, while standard PLMs struggle at random guess levels (see Figure 7 and Appendix D). This demonstrates forgetting PLMs' ability to efficiently adapt to new languages, particularly dissimilar ones, in low-data settings.

---

[2]Precisely, it simulates vocabulary swappings, causing drastic changes to the input of the body.

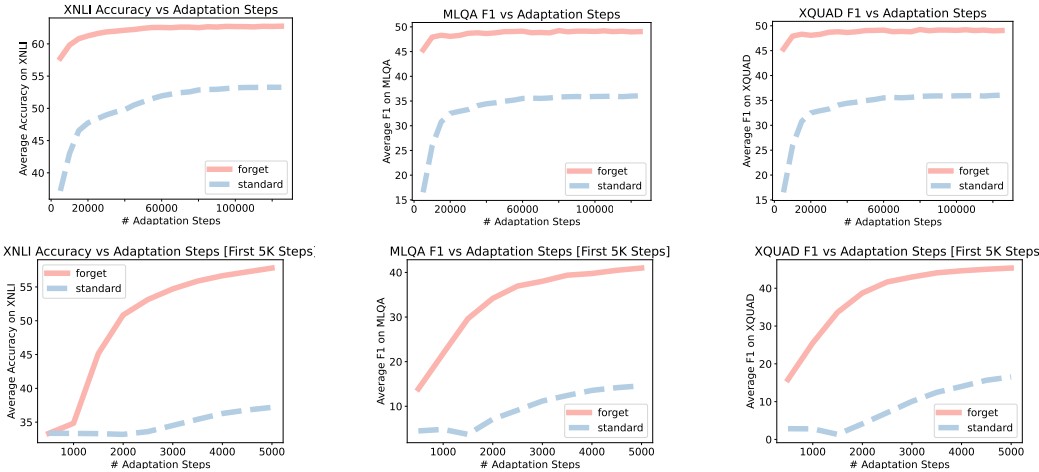

Figure 5: Adaptation curves on XNLI, MLQA, and XQuAD. Numbers aggregated across languages. The first row contains the full adaptation curves, which comprises 125K adaptation steps. The second row contains the zoom-in versions of curves for the first 5K adaptation steps. Forgetting PLMs converge faster than standard PLMs; for instance, on XQuAD (the last plot), forgetting PLMs reach 92% of their final performance within 5K updates, while standard PLMs only reached 53% of their final performance at that point.

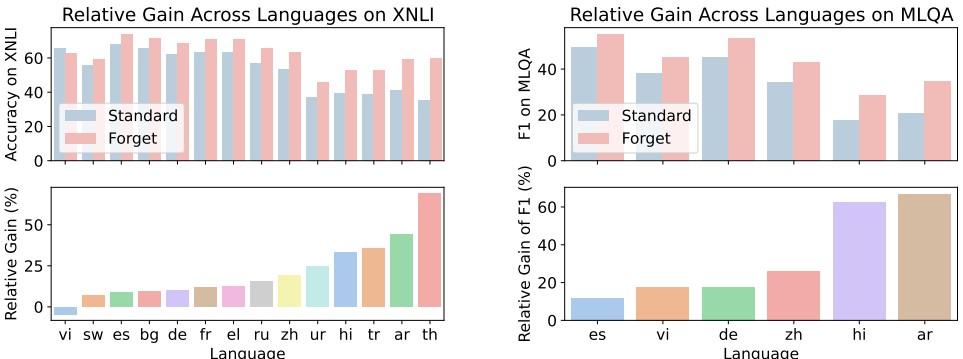

Figure 6: Relative gains of forgetting PLMs over standard PLMs across languages. Forgetting yields substantial relative gains for languages like Arabic, Hindi, Thai, Turkish, and Urdu. However, for languages closely related to English, such as German, the relative gains from forgetting are modest.

## 5 Related Work

### 5.1 Forgetting and its Positive Roles

The common perception of forgetting is that it implies weak memory and a loss of acquired knowledge, thus it is often regarded as a sign of *un-intelligence* or an undesirable property. In neural networks, *catastrophic forgetting* [McCloskey and Cohen, 1989, Ratcliff, 1990, Kirkpatrick et al., 2017] is portrayed as a forgetting phenomenon where neural networks lose the ability to predict old patterns after new inputs alter their weights. Forgetting in this context has negative consequences, as the new knowledge overwrites the old. Plenty of prior research strives to overcome catastrophic forgetting and enable continual learning [Schmidhuber, 2013, Kirkpatrick et al., 2017, Lopez-Paz and Ranzato, 2017, Shin et al., 2017, Schwarz et al., 2018, Mallya and Lazebnik, 2018, Parisi et al., 2019, Rolnick et al., 2019, Beaulieu et al., 2020, Veniat et al., 2020, Gaya et al., 2023, Khetarpal et al., 2022].

Our work differs from the above ones in that our subject is *intentional forgetting* rather than passive forgetting and its associated negative impact. To put it in another way, we seek to understand how

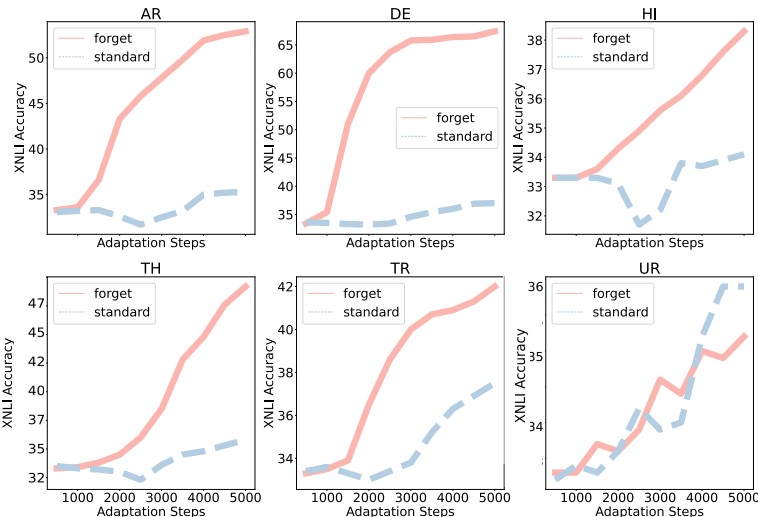

Figure 7: Adaptation curves on XNLI within 5K updates for individual languages: Bulgaria, Greek, Spanish, French, Russian, Swahili, Vietnamese and Chinese. For all languages except Urdu, the forgetting PLMs converge faster than the standard PLMs during the language adaptation stage.

forgetting – if purposely incorporated as an active process during training – might *help* new learning. Similar positive roles of forgetting have been discussed in the literature. Specifically, Pastötter et al. [2008] demonstrate forgetting enhances the learning of new information by resetting the encoding process and holding the attention at high levels; Levy et al. [2007] show that it helps second language acquisition by inhibiting the native language; Barrett and Zollman [2009] find it promote the emergence of an optimal language by preventing partial success from reinforce sub-optimal practice. Nørby [2015] further suggests forgetting serves adaptive functions, helping people regulate emotions, acquiring knowledge and staying attuned to the context. More recently Anderson and Hulbert [2021] reviews evidence on active forgetting by prefrontal control and shows how it can adapt the memory to suit either emotional or cognitive goals.

## 5.2 Forgetting Via Partial Neural Weights Reset

In neural networks, forgetting can be instantiated in many forms. A simple way is to reset subsets of parameters before the next round of learning. Iterations of such resetting have been shown to benefit generalization with low compute and low data for computer vision tasks [Frankle and Carbin, 2019, Alabdulmohsin et al., 2021, Taha et al., 2021, Ramkumar et al., 2023]. More recently, Zhou et al. [2022] demonstrate a similar forgetting strategy helps image classification and language emergence. Closely linked to our method, Chen et al. [2022] forget node embeddings in order to truncate infinite message-passing among nodes and thereby aid new graph reasoning with new nodes. Our work uses similar forgetting mechanism over token embeddings, improving new language reasoning with new tokens. As far as we know, *we are the first to bring forgetting into pretraining and demonstrate that forgetting pretraining boosts linguistic plasticity*. A relevant thread in reinforcement learning (RL) research studies the plasticity loss phenomenon [Lyle et al., 2023, Nikishin et al., 2023]. Recent work explores similar forgetting approaches to improve plasticity. Igl et al. [2021] periodically reset the current policy by distilling it into a reinitialized network throughout training. Intuitively, this releases network capacity storing suboptimal policies and opens up space for the the yet-to-be-discovered optimal (final) policy. Simpler methods just reset an agent's last layers [Nikishin et al., 2022], preventing overfitting to early experiences and *primacy bias*. Resetting parameters also improves sample efficiency by allowing more updates per environment interaction [D'Oro et al., 2023].

## 5.3 Making Pretrained Language Models Multilingual

Pretraining on multilingual data makes PLMs multilingual [Conneau et al., 2020] but has downsides like needing large multilingual corpus with appropriate mixing, potential interference among lan-

guages, and difficulty of covering all languages. Alternatively, the line of research on cross-lingual transfer makes PLMs multilingual by extending English-only PLMs to other languages. Artetxe et al. [2020] demonstrate possibility of such extension by relearning the embedding layer with unsupervised data from the new language. Marchisio et al. [2022] further increase computational efficiency using a mini-model proxy. Liu et al. [2023a] use a similar partial reset-reinit approach in vision-language settings. Approaches based on adapters and sparse finetuning have also been proposed [Pfeiffer et al., 2020, 2022, 2021, Ansell et al., 2022]. Adapters are bottleneck layers (usually placed after the feedforward layers) that add extra capacity when adapting to a different task or language. Our proposed forgetting mechanism can be applied to adapter-based methods as we can allow forgetting to happen in the adapter layers. The current choice of forgetting embeddings keeps the architecture intact and incurs no additional hyperparameter tuning, allowing us to understand the fundamental capability of forgetting pretraining.

# 6 Conclusion & Future work

## 6.1 Conclusions

While forgetting is usually perceived as negative, recent work points out that it can also be beneficial in certain cases, particularly for quickly learning new tasks, training in non-stationary environments [Igl et al., 2021, Nikishin et al., 2022, D'Oro et al., 2023] and improving sample efficiency [Taha et al., 2021, Zhou et al., 2022]. Joining this line of work, our paper demonstrates that forgetting techniques can improve pretrained language models by imbuing them with more linguistic plasticity. Specifically, our proposed *active forgetting* mechanism can create PLMs that are easier to rewire for new lingual spaces. Experiments with RoBERTa show that models pretrained via active forgetting can better learn from small amounts of data while also enjoying faster convergence during language adaptation, particularly for languages distant from English.

Going beyond language adaptation, we argue that PLMs with more plasticity are a promising direction for future research, as they allow easier adaptation to various tasks, domains, languages and can evolve faster as the real world changes. Unlike symbolic methods, such as knowledge graphs, which can easily rewire a fact by modifying the corresponding knowledge triplet, current static PLMs are harder to rewire since changing one fact via updating model weights may disrupt multiple other facts without substantial post-hoc intervention. Improving the rewirability via forgetting pretraining thus can be seen as one way of imbuing PLMs with similar benefits as symbolic methods (making the resulted model more controllable i.e. can be modified with tiny cost), complementing the line of post-hoc model editing research [Mitchell et al., 2021, 2022].

## 6.2 Limitations

Our work uses the simplest forgetting approach - directly resetting embeddings to random initialization. Advanced techniques like gradually injecting noise could be explored. We focus on masked language modeling pretraining with language-specific tokenizers. Applying active forgetting to auto-regressive LMs, other pretraining methods (e.g. deberta pretraining [He et al., 2021b,a]), and various tokenization is promising future work. More broadly, current large language models need more plasticity to expand across tools, tasks, and domains. Our work takes an initial step, showing directly resetting embeddings can significantly improve model plasticity. Further research on more sophisticated forgetting techniques during pretraining could unlock additional gains.

On the theory front, potential connections can be made between forgetting and meta-learning [Schaul and Schmidhuber, 2010, Thrun and Pratt, 2012, Andrychowicz et al., 2016, Finn et al., 2017] since both attempt to learn solutions that can quickly adapt themselves to new inputs. Another possible theoretical explanation for why active forgetting works so well might be related to the flatness of the solution in the loss landscape [Alabdulmohsin et al., 2021]. Flatter minima tend to enjoy better generalization [Liu et al., 2023b]. Thus, it might be worthwhile to study the flatness of the transformer body during the forgetting pretraining.

## Acknowledgments and Disclosure of Funding

We would like to thank our reviewers for their valuable suggestions. We are also grateful to those who engaged in interesting discussions during the project, including Pasquale Minervini, Xuanli He, Jiayi Wang, Yuxiang Wu, Hila Gonen, Dieuwke Hupkes, Fabio Petroni, Naila Murray, Alexis Thual, Nicola Cancedda, Yingchen Xu, and Hubert Jacob Banville. Yihong would like to express her thanks to the FAIR-UCL PhD program for generously funding her PhD. David Adelani acknowledges the support of DeepMind Academic Fellowship programme.

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
