_{\text{emb}}^{(n)}, P_{\text{body}}^{(n)}\}, O^{(n)} = \{O_{\text{emb}}^{(n)}, O_{\text{body}}^{(n)}\}$.

1: $n_{\text{emb}} = n \bmod K$
2: $\alpha_{\text{body}} = s(n_{\text{body}})$ {adjust learning rate based on $n$}
3: $\alpha_{\text{emb}} = s(n_{\text{emb}})$
4: $G^{(n)} = f(P^{(n-1)}, \cdot)$ {compute all gradients}
5: $P_{\text{body}}^{(n)}, o_{\text{body}}^{(n)} = g(G_{\text{body}}^{(n)}, P_{\text{body}}^{(n-1)}, o_{\text{body}}^{(n-1)}, \alpha_{\text{body}}, n)$ {update the transformer body}
6: **if** $n_{\text{emb}} == 0$ **then**
7: $\quad P_{\text{emb}}^{(n)} = \Theta$ {periodical reset token embeddings and the relevant optimizer states}
8: $\quad o_{\text{emb}}^{(n-1)} = 0$
9: **end if**
10: $P_{\text{emb}}^{(n)}, o_{\text{emb}}^{(n)} = g(G_{\text{emb}}^{(n)}, P_{\text{emb}}^{(n-1)}, o_{\text{emb}}^{(n-1)}, \

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

| Name | Code | Family | Script | Morphology |
|------|------|--------|--------|------------|
| Arabic | ar | Semitic | Arabic (Abjad) | Introflexive |
| Bulgaria | bg | IE:Balto-Slavic | Cyrillic | Analytic |
| German | de | IE:Germanic | Latin | Fusional |
| Greek | el | IE:Hellenic | Greek | Fusional |
| English | en | IE:Germanic | Latin | Analytic |
| French | fr | IE:Romance | Latin | Fusional |
| Hindi | hi | IE:Indo-Iranian | Devanagari | Fusional |
| Russian | ru | IE:Balto-Slavic | Cyrillic | Fusional |
| Spanish | es | IE:Romance | Latin | Fusional |
| Swahili | sw | Niger-Congo:Bantu | Latin | Agglutinative |
| Thai | th | Tai-Kadai | Thai | Analytic |
| Turkish | tr | Turkic | Latin | Agglutinative |
| Urdu | ur | IE:Indo-Iranian | Perso-Arabic | Fusional |
| Vietnamese | vi | Austroasiatic | Latin | Analytic |
| Chinese | zh | Sino-Tibetan | Chinese | Analytic |

Table 5: Languages by family, script, and morphology.

# A  Discussion on Experimental Setup A Low-Data Regime

A common experimental setup for adapting to a target language is to use all the available data in that language from sources such as Wikipedia [Artetxe et al., 2020, Ansell et al., 2022] and CC100 [Marchisio et al., 2022]. In this setup, the numbers of tokens typically used for adapting each language might differ greatly, ranging from 13.9M to 70.5B, as summarized via Table 5.

Our work, however, investigates a different setup where we control the adaptation data to 5 million tokens for each language. This can be highly relevant when studying generalisation to completely new languages, which require expanding the vocabulary. We acknowledge that dealing with real-world low-resources languages can be more challenging than such low-data setup used. And there are already rich work addressing low-resource issues: multilingual pretraining [Conneau et al., 2020, Pfeiffer et al., 2022], multilingual adapters [Pfeiffer et al., 2020, Ansell et al., 2022], multilingual adaptation [Tang et al., 2020, Alabi et al., 2022], and multilingual regularization [Pfeiffer et al., 2021]. Nevertheless, we would like to highlight the importance of our low-data regime. The challenge of "low-resource" involve multiple *entangled factors*: the quality of the tokeniser, the amount of data, whether the script/language family is distant to the pretraining language(s) etc. Simulating a low-data regime allows us to control these factors and isolate the effects of the factor that we are interested in – the amount of data in the new language. This factor is essential to our work as our research goal is plasticity i.e. rewiring model prediction with as little new information as possible. Simulating various amount of data in the new language allows us to compare model plasticity as shown in Figure 3, and thus contribute a clean piece of knowledge in the line of plasticity research [Lyle et al., 2023, Nikishin et al., 2023].

# B  Discussion on Experimental Framework Choice for Studying Pretrained Language Models' Plasticity

Our motivation is to improve language models' plasticity. Plasticity of neural networks have been studied in graph learning, computer vision and reinforcement learning [Taha et al., 2021, Chen et al., 2022, Lyle et al., 2023, Nikishin et al., 2023], where forgetting-relearn methods show promise. Our goal is to study plasticity in the context of pretrained language models. We believe this is a emerging research direction and will thrive in the following years. However, translating the plasticity concept to the language model setting is not trivial due to the lack of clear experimental setups. We note that, despite the model differences, almost all language models begins with a token embedding layer. As often tied to a specific vocabulary, the token embedding layer limits the plasticity, preventing generalisation to a new vocabulary. This observation inspires us to explore the plasticity of language models by manipulating the token embedding layer. Artetxe et al. [2020] draws our attention as it offers a nice experimental framework of only manipulating the token embedding layer for adapting between languages.

| Language | CC-100 | Wikipedia |
|---|---|---|
| sw | 345M | 13.9M |
| ur | 832M | 41.5M |
| hi | 2.13B | 54.6M |
| ar | 4.15B | 337M |
| tr | 4.19B | 157M |
| th | 6.09B | 70.7M |
| el | 6.1B | 148M |
| bg | 7.9B | 134M |
| zh | 9.72B | 584M |
| es | 11.6B | 1.17B |
| fr | 13.3B | 1.71B |
| de | 14.1B | 2B |
| vi | 28.9B | 300M |
| ru | 34.9B | 1.25B |
| **en** | 70.5**B** | 4.25**B** |

Table 6: Numbers of tokens for different languages on the two multilingual corpus, CC-100 and Wikipedia, in ascending order of CC100. The English one is used as pretraining corpus while the others are used for language adaptation.

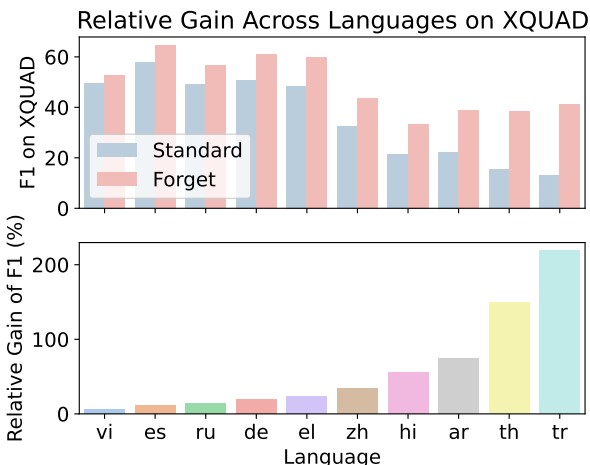

Figure 8: Relative gains of forgetting PLMs over standard PLMs across languages on XQuAD. Languages that are less related to the pretraining language (English), such as Turkish, Thai, Arabic, Hindi, benefit more from forgetting PLMs.

## C  More Results on Distant Languages Benefits More From Forgetting

We are interested in how forgetting PLMs can improve adaptation to different languages. We compare the results of various languages on three benchmarks: XNLI, MLQA and XQuAD. We use Figure 6 and Figure 8 to illustrate the relative gains from active forgetting on each benchmark. We find that languages that are less related to the pretraining language, which in this case is English, benefit more from forgetting PLMs.

## D  More Results on Forgetting PLMs Converge Faster In The Language Adaptation Stage

Figure 9 displays the adaptation curves for several languages (Arabic, German, Hindi, Thai, Turkish, and Urdu) during full training runs of 125,000 steps. This complements Figure 7, which focuses

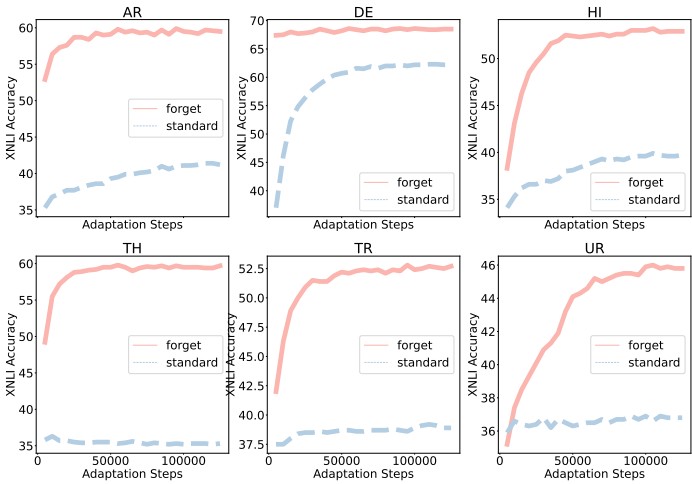

Figure 9: Adaptation curves on XNLI for individual languages: Arabic, German, Hindi, Thai, Turkish, and Urdu. Forgetting helps more languages that are distant to English (the pretraining language).

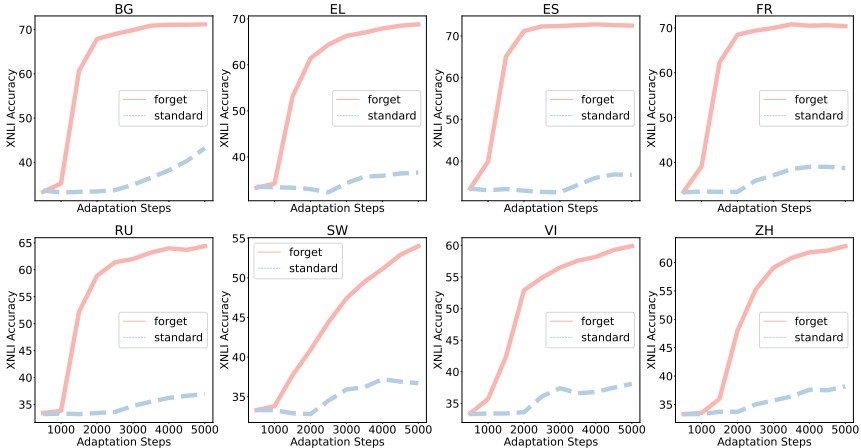

Figure 10: Adaptation curves on XNLI within 5K updates for individual languages: Bulgaria, Greek, Spanish, French, Russian, Swahili, Vietnamese and Chinese. Forgetting PLMs converge faster than standard PLMs.

on the first 5,000 steps. Similar convergence patterns can be observed for additional languages, as shown in Figure 10 and Figure 11.

## E  More Detailed Analysis

### E.1  Impact of Forgetting Frequency

We would like to elaborate on our choice of forgetting frequency $K$. In our preliminary experiments, we tried $K = 100, 1000, 5000$. We find $K = 1000$ works well and thus sticks with it. Since we don't want to overtune the hyperparameters, we just use the same $K$ for all the experiments. We include the loss curves of $K = 100$ and $K = 5000$ here. We can see that both forgetting too frequently and forgetting too infrequently will hurt the performance. Too frequent forgetting leaves little time for the

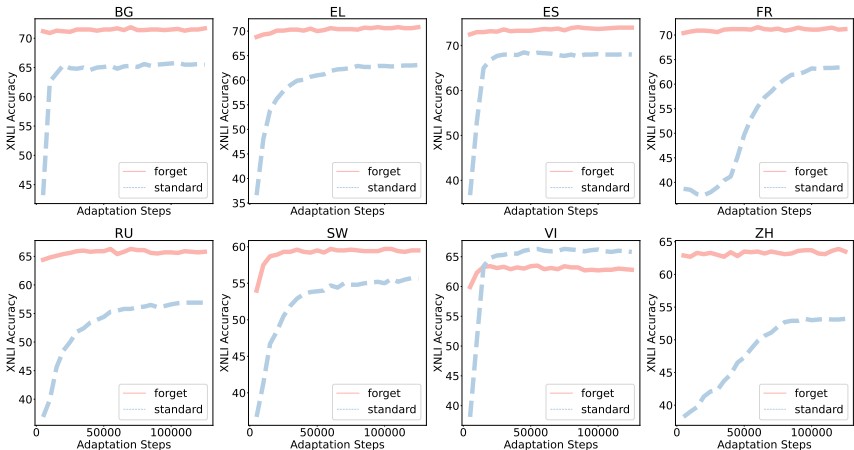

Figure 11: Adaptation curves on XNLI for individual languages: Bulgaria, Greek, Spanish, French, Russian, Swahili, Vietnamese and Chinese. Across all the languages except on Vietnamese, the forgetting PLMs reach a better performance level than their standard counterparts.

body to learn something meaningful (the pretraining loss stuck around 11). Too sparse forgetting will make the body hard to adjust to the next forgetting, causing divergence as pretraining goes on.

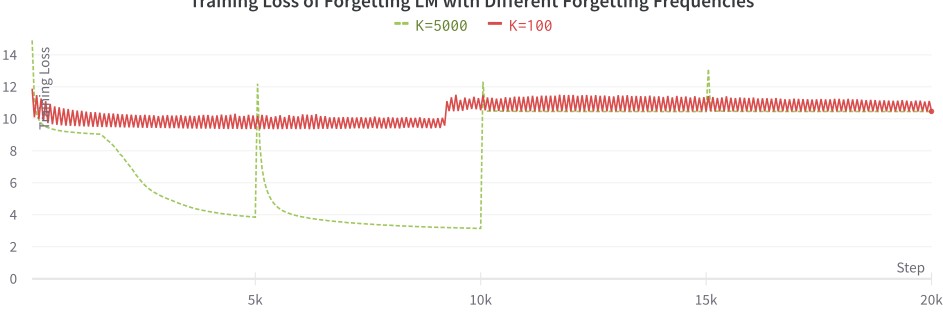

Figure 12: Impact of Forgetting Frequency.

## E.2 Multilingual Pretraining and Forgetting

Our work aim to have a flexible language model by pretraining with forgetting. No matter the pretraining corpus is monolingual or multilingual, this language model should easily generalise itself to unseen languages. This is different from the scenario of multilingual PLMs like XLM-R [Conneau et al., 2020], which requires seeing all the data for all languages from the scratch. Once done with pretraining and there is some new language distant from the pretraining languages to support, the multilingual PLMs might still struggle with zero-shot transfer as shown in several low-resources language research [Ebrahimi et al., 2022, Adelani et al., 2021, 2022].

Nevertheless, we ran additional experiments on multilingual pretraining with forgetting. For a fair comparion, we trained a multilingual RoBERTa-base of the same model size as our monolingual model. Language Emb/Task Body Adaptation refers to separately adapting embeddings with 5M tokens of Thai unlabelled data and adapting body with English NLI data. Task Full Model Adaptation refers to adapting the full model with English NLI data. Note that Thai is already included in multilingual CC100 (6B tokens in the original dataset, 720M tokens in our subsampled dataset). We measure the zero-shot Thai XNLI Accuracy.

| Pretrain Corpus | #Langs | #Params | Pretraining Method | Adaptation Framework | Acc |
|---|---|---|---|---|---|
| 300GB English | 1 | 125M | Standard-RoBERTa (base) | Lang Emb/Task Body | 35.3 |
| 300GB English | 1 | 125M | Forget-RoBERTa (base) | Lang Emb/Task Body | 59.7 |
| 50GB Multilingual | 100 | 125M | Standard-RoBERTa (base) | Lang Emb/Task Body | 49.4 |
| 50GB Multilingual | 100 | 125M | Forget-RoBERTa (base) | Lang Emb/Task Body | 55.0 |
| 50GB Multilingual | 100 | 125M | Standard-RoBERTa (base) | Task Full Model | 60.0 |
| 2.5TB Multilingual | 100 | 270M | XLM-RoBERTa (base) | Task Full Model | 72.4 |

We can see that multilingual pretraining indeed helps cross-lingual transfer when the language is in the pretraining data. On the other hand, we can also observe that forgetting indeed lifts the adaptation performance:

- Comparing Row 3 and Row 4 (49.4 vs 55.0), we can see that, forgetting also helps adapt multilingual pretrained models.

- Comparing Row 1 and Row 2 (35.3 vs 59.7), we can see that forgetting helps monolingual pretrained models a lot

- XLM-R (base) outperform best our multilingual pretrained baselines (72.4 vs 60.0). This is no surprise due to its large pretraining corpus (10x our multilingual corpora) and model size (2x our multilingual model).

### E.3 Full Model Task Adaptation and Forgetting

The language/task adaptation does not use any labelled data. It only uses the unlabelled data from the new language. In contrast, Standard adaptation relies on labelled data, which is expensive for a new downstream language. In our case, we only found Arabic NLI data. The amount of labelled data is not enough to adapt an English NLI model to Arabic NLI without proper regularization.

Our experimental setup follows Artetxe et al. [2020], where standard-pretraining + language/task adaptation (MonoTrans) is shown to be competitive among a few baselines for zero-shot unsupervised cross-lingual transfer. On top of this finding, our proposed forgetting method can further improve the sample-efficiency of the language/task adaptation, reducing the amount of unsupervised data needed for the new language. This is motivated by a practical scenario where the new languages contain only several thousands of tokens to a few millions of tokens (e.g. the corpus for the new language might contain only 2-3 books).

| Method | Supervised Data | Unsupervised Data | Acc |
|---|---|---|---|
| standard pretraining + standard adapt | 6.7K | 0 | 32.8 |
| standard pretraining + language/task adaptation | 0 | 5M | 41.2 |
| forget pretraining + standard adaptation | 6.7K | 0 | 34.2 |
| forget pretraining + language/task adaptation | 0 | 5M | 59.7 |

### E.4 Impact of Adaptation Data Amount

Evidence of high sample efficiency can be found by comparing the performance drop of standard PLMs and forgetting PLMs when the adaptation data change from a high-data setting [Artetxe et al., 2020, Marchisio et al., 2022] to a low-data setting that our work is considering.

| Method | Avg adaptation #tokens | Avg XNLI performance |
|---|---|---|
| Standard (Kelly et al 2022 [27]) | 10.3B | 72 |
| Standard (Artetxe et al 2022 [18]) | 569M | 66.7 |
| Standard | 5M | 53.3 |
| Forgetting | 5M | 62.7 |