# OpenReview forum: "Improving Language Plasticity via Pretraining with Active Forgetting"
_NeurIPS.cc/2023/Conference — NeurIPS 2023 poster_

### Official Review · Reviewer_LV8K · 2023-06-27

**Soundness:** 4 excellent
**Presentation:** 3 good
**Contribution:** 3 good
**Rating:** 7
**Confidence:** 4

**Summary:**

This paper proposes active forgetting, a rather straightforward method that resets the embedding layer every K updates during pretraining, to quickly adapt PLMs to new languages. Through experiments on different language pairs with RoBERTa, the authors claim that the proposed method can induce faster convergence and better performance when the languages are distant from English.

**Strengths:**

- This paper is well-written in general, with a clear motivation and some novelty.

- It is appreciated that the author conduct experiments on many languages from similar languages like German to distant languages like Thai.

- The method itself is counter-intuitive, as it resets the embedding layer periodically which intuitively can be bad for PLMs, but effective to improve the cross-lingual transfer.

- The authors give in-depth analysis and insights into their proposed method, which I believe is interesting to the community.


**Weaknesses:**

Some figures are inconsistent and visually not good. For example, the upper three subfigures in Figure 4 have a visually bad axis scale. Some values are rounded to integers while some are not, as in Figure 6. The authors should carefully redraw and prettify them in the camera-ready version.

**Questions:**

- In the abstract, the paper claims that it is data and compute inefficient to learn a new embedding layer. But this paper does not address this problem, since a new embedding layer for new languages still has to be learned. Therefore I don't think it is necessary to mention that in the abstract.

- The paper claims the method ensures high sample efficiency. How does the experiment support this?

- Forgetting is “generally” a bad thing for PLMs, but this work counter-intuitively shows that active forgetting can be beneficial for cross-lingual transfer. The authors are encouraged to give more intuition or explore the reasons behind this phenomenon, in a cross-lingual perspective.

- For the experiment w.r.t. RQ3, the authors show that active forgetting is particularly helpful when the new languages are (typological) different from the pretraining language. However, I see also an important possible influencer: the script. The authors are encouraged to also explore if scripts can influence the transfer performance under active forgetting.

- The frequency of active forgetting is set to K=1000. As this is an important hyperparameter, I would encourage the authors to justify their choice.

- I would be very interested to see a plot of loss over updates during language adapting. I would expect the loss to go down in general but fluctuate a lot every time when active forgetting is used.

- It’s natural to think that this active forgetting could be used in other parts of the models instead of merely on embeddings. Therefore it would be very interesting to explore: active forgetting on which part of the models is the most effective?


**Limitations:**

The authors include a section of Limitations in the paper, where they mention that this paper only focuses on the simplest forgetting, which is only applied to the embeddings.

---

> ### Author Rebuttal · Authors · 2023-08-10
>
> Thank you for your valuable comments.
>
> ### We appreciate that you recognise the soundness and contribution of our work. We would like to address your comments as follows.
>
> **Figures 4 and 6**: Thank you for pointing out the issues. We will polish the figures in the camera-ready version.
>
> **Q1**:
> > In the abstract, the paper claims that it is data and compute inefficient to learn a new embedding layer. But this paper does not address this problem, since a new embedding layer for new languages still has to be learned.
>
> We would like to clarify the confusion here. Although the new embedding layer still has to be learned, forgetting PLMs require less compute and data in the new language to reach a good performance:
> - 1) less compute. Fig 4 in our paper shows that forgetting PLMs converge fast within $5K$ adaptation steps. So we can actually stop the adaptation training earlier than the standard methods. For example, on XQUAD, in order to converge to $90\%$ of their final performances, forgetting PLMs take only ~$5K$ steps while standard PLMs take ~$20K$ steps. This leads to **4x** speedup for the language adaptation run.
> - 2) less adaptation data. We did a few ablations on the adaptation data quantity and studied its impact on learning a new embedding layer. Fig 3 in our paper summarizes the results for learning a new English embedding layer. We can see that, when adaptation data is less than $10M$ (often true for low-resources languages), forgetting PLMs consistently outperform standard PLMs. Thus, to reach the same level of performance in low-resources setting, forgetting PLMs require less adaptation data while relearning the English embeddings (an extreme example would be in Fig 3 in our main paper, to reach 40 on NLI, standard PLMs require ~$100K$ adaptation data while forgetting PLMs only require ~$5K$).
>
> **Q2**:
> > The paper claims the method ensures high sample efficiency. How does the experiment support this?
>
> The second point in our response to Q1 addresses this question. More evidence can be found by comparing the perfomance drop of standard PLMs and forgetting PLMs when the adaptation data change from a high-data setting [18,27] to a low-data setting that our work is considering:
>
> | Method                  | Avg adaptation #tokens | Avg XNLI performance |
> | :---------------------- | ---------------------: | -------------------: |
> | Kelly et al 2022 [27]   |                  10.3B |                   72 |
> | Artetxe et al 2022 [18] |                   569M |                 66.7 |
> | Standard                |                     5M |                 53.3 |
> | Forgetting              |                     5M |                 62.7 |
>
> We can see that, when the adaptation data is reduced from 569M to 5M, forgetting only drops about 6% (from 66.7 to 62.7) while standard drops about 20% (from 66.7 to 53.3). When the adaptation data amount drops, forgetting PLMs still retain a relatively good performance compared to standard PLMs.
>
>
> **Q3**:
> > The authors are encouraged to give more intuition or explore the reasons behind this phenomenon, in a cross-lingual perspective.
>
> Our intuition is that the periodic forgetting of the token embedding layer will force the transformer body learn better high-level abstraction. Because every time forget happens, the body kind of "re-derive" these abstraction; by repeating this process again and again, the body finally learns to abstract instead of short-cut memorizing particular embedding values. During cross-lingual transfer, a body with more high-level abstraction can be more easily transferred to new languages, since the high-level abstraction is more language-agnostic.
>
> Our intuition can also be supported by cognitive science literature (see sec 5.1 in our paper) where forgetting is shown beneficial for learning to abstract [27,29] and new languages[28].
>
> We will add more discussion on this in the camera-ready version.
>
> **Q4**:
> >  For the experiment w.r.t. RQ3, the authors show that active forgetting is particularly helpful when the new languages are (typological) different from the pretraining language. However, I see also an important possible influencer: the script.
>
> This indeed a very valuable comment. We did observe some impact of the script. For example, although Vietnamese and Swahili are both distant from English, yet forgetting brings modest or no improvements. We guess it might be because they are written in the Latin script, which is the same as the pretaining English. We will add this discussion in the camera-ready version.
>
> **Q5**: Frequency of Forgetting
>
> Please see general response.
>
> **Q6**: Loss Curves of Forgetting
>
> Yes, the forgetting indeed creates a spike and then the model learn to recover to normal loss. Please see general response for details.
>
> **Q7**:
> > active forgetting on which part of the models is the most effective?
>
> We are also excited about this direction. Our guess is that it will depend on the task. Token embeddings play a big role in cross-lingual transfer, so forgetting on token embeddings is effective. But for other tasks, forgetting on other parts of the model might be more effective. We will explore this in the future.
>
> ------
>
> References 1-26 can be found in our responses to the other reviewers.
>
>  [27] Marchisio, Kelly et al. Mini-model adaptation: Efficiently extending pretrained models to new languages via aligned shallow training. ACL 2023 Findings
>
>  [28] Levy, Benjamin J et al. Inhibiting your native language: The role of retrieval-induced forgetting during second-language acquisition. Psychological Science, 2007
>
>  [29] Simon Nørby. Why forget? on the adaptive value of memory loss. Perspectives on Psychological Science, 2015
>
>  [30] Michael C. Anderson and Justin C. Hulbert. Active forgetting: Adaptation of memory by prefrontal control. 2022

---

> > ### Comment · Reviewer_LV8K · 2023-08-18
> >
> > Thank you very much for your detailed response and all my questions have been answered now.

---

### Official Review · Reviewer_1FBf · 2023-06-30

**Soundness:** 3 good
**Presentation:** 3 good
**Contribution:** 2 fair
**Rating:** 6
**Confidence:** 4

**Summary:**

The paper proposed an active forgetting mechanism for PLMs pre-training for cross-lingual transfers and adaptations. The authors propose a multi-stage adaptation framework for better cross-lingual transfer/adaption: 1) First, by resetting embedding layers every K update during pre-training for monolingual RoBERTa model; 2) Second, by adapting parameters separately (embeddings vs backbone) for language-specific knowledge and task-specific knowledge (language+task adaptation). The paper shows the proposed mechanism helps faster convergence during cross-lingual transfers and adaptation as well as achieving better downstream task performances.


**Strengths:**

* The paper shows an interesting empirical finding that resetting partial parameters (in this case embeddings) during pre-training in English leads to better transfers/adaptation in cross-lingual settings.
* The paper is clear-written and the proposed method is effective empirically for cross-lingual transfers.


**Weaknesses:**

- Although the proposed method achieved good performances in downstream tasks, this work lacks proper ablation experiments on showing the gain is from resetting during pre-training, vs the gain of adapting/resetting parameters separately during the language+task adaptation stage. I.e. the paper should include the results of the following 4 types of experiments:
    * Standard pre-training + standard adaptation
    * Forgetting pre-training + standard adaptation
    * Standard pre-training + language/task adaption
    * Forgetting pre-training + language/task adaption

- Practically, efficiency (parameters and training) could be an issue with the language adaptation + task adaptation issue.

**Questions:**

* See weaknesses.
* Have you tried resetting other parameters during pre-training? What about other PLMs? e.g. Does deBERT style training affect the results?

* References:

     1. Chen Liu, Jonas Pfeiffer, Anna Korhonen, Ivan Vulić, and Iryna Gurevych. Delving Deeper into Cross-lingual Visual Question Answering. Findings of EACL, 2023 → Partial resetting, and re-initialization of parameters for cross-lingual generalization in VL setting.

     2. Vijaya Raghavan T Ramkumar, Elahe Arani, Bahram Zonooz, Learn, Unlearn and Relearn: An Online Learning Paradigm for Deep Neural Networks, TMLR, 2023 → Selective parameters resetting for continual learning (online and few-shot).

---

> ### Author Rebuttal · Authors · 2023-08-09
>
> Thank you for your review. We appreciate that you recognise the effectiveness of our method and find our finding interesting. We would like to address your concerns as follows.
> ### First, we want to elaborate on our experimental setup.
> > this work lacks proper ablation experiments on showing the gain is from resetting during pre-training, vs the gain of adapting/resetting parameters separately during the language+task adaptation stage. ...
>
> We run quick experiments on Arabic and include numbers for standard adaptation here. However, we also want to highlight the difference between standard adaptation and language/task adaptation.
>
> Standard adaptation relies on **labelled** data, which is expensive for a new downstream language. In contrast, the language/task adaptation **does not use any labelled data**. It only uses the unlabelled data from the new language. In our case, we only found $6.7K$ Arabic NLI data. The amount of labelled data is not enough to adapt an English NLI model to Arabic NLI without proper regularization.
>
> Our experimental setup follows [18], where standard-pretraining + language/task adaptation (MonoTrans) is shown to be competitive among a few baselines for zero-shot unsupervised cross-lingual transfer. On top of this finding, our proposed forgetting method can further improve the sample-efficiency of the language/task adaptation, reducing the amount of unsupervised data needed for the new language. This is motivated by a practical scenario where the new languages contain only several thousands of tokens to a few millions of tokens (e.g. the corpus for the new language might contain only 2-3 books).
>
>
>   **Method **                                          | **Supervised Data** | **Unsupervised Data** | **Arabic XNLI ACC**
> ------------------------------------------------------|---------------------|-----------------------|---------------------
>  **standard pretraining + standard adapt**            | 6.7K                | 0                     | 32.8
>  **standard pretraining + language/task adaptatioon** | 0                   | 5M                    | 41.2
>  **forget pretraining + standard adaptation**         | 6.7K                | 0                     | 34.2
>  **forget pretraining + language/task adaptation**    | 0                   | 5M                    | 59.7
>
>
> ### Second, we would like to address your questions.
> **Q1**:
> > Have you tried resetting other parameters during pre-training?
>
> We did try resetting the bias terms in the language model head, though it didn't help much in our preliminary experiments.
>
> We decided to focus on token embedding layer as prior work [18-24] demonstrate that the token embedding layer, which captures most lexical meanings, is crucial for cross-lingual transfer. Moreover, since the token embedding layer is not only the first layer but also the last layer (RoBERTa and many other language models use tied token embeddings), resetting the token embedding layer is equivalent to resetting the last layer. This echos the findings in [4,5,7,8,10], where resetting **later** layers is more effective to model plasticity.
>
> In the future, it could be interesting to try out resetting other parts and see if they benefit particular tasks. For example, we can leverage some outcome from the line of interpretable LM. Say if we understand the functions of a particular subnetwork, we can consider selectively forgetting them to help relevant downstream tasks, achieving similar effects as parameter-efficient tuning methods[25-26].
>
> **Q2**:
> > What about other PLMs? e.g. Does deBERT style training affect the results?
>
> We would like to extend our experiments to more pretrained models. deBERT sounds like a good candidate. However, we are currently bottlenecked by the computational resources, similar as many other pretraining research. One compiling of the entire experimental pipeline requires at least 2 pretraining runs, 48 language adaptation runs, 6 task adaptation runs, which takes about 38460 GPU hours (V100 32GB). We are working hard to extend to more models but would require more time and computational resources.
>
> On the other hand, we chose RoBERTa as it is one of the most widely-used open-sourced pretrained language models. The effectiveness of our method on RoBERTa shows the potential of our method for the transformer family, as our method is not tied to a particular RoBERTa component but generalizable to any language model with a token embedding layer. We will open-source our code to facilitate future research on other pretrained models.
>
> **References**:
> Thank you for the references. We will add them to our camera-ready version.
>
> -----
>
> References 1-14 can be found in our response to Reviewer ar7w. References 15-17 can be found in our response to Reviewer TTi1
>
> [18] Artetxe, Mikel et al. "On the Cross-lingual Transferability of Monolingual Representations." ACL 2020.
>
> [19] Minixhofer, Benjamin,  et al. "WECHSEL: Effective initialization of subword embeddings for cross-lingual transfer of monolingual language models." NACCL 2022.
>
> [20] Dobler, Konstantin et al. "FOCUS: Effective Embedding Initialization for Specializing Pretrained Multilingual Models on a Single Language." arXiv preprint arXiv:2305.14481 (2023).
>
> [21] Tran, Ke. "From English to Foreign Languages: Transferring Pre-trained Language Models." (2019).
>
> [22] Jain, Neel, et al. "How to Do a Vocab Swap? A Study of Embedding Replacement for Pre-trained Transformers." (2022).
>
> [23] Barret Zoph, Deniz Yuret, Jonathan May, and Kevin Knight. 2016. Transfer Learning for Low-Resource Neural Machine Translation. EMNLP 2016
>
> [24] Chao Xing, el al. Normalized Word Embedding and Orthogonal Transform for Bilingual Word Translation. NAACL 2015.
>
> [25] Hu, Edward J., et al. "LoRA: Low-Rank Adaptation of Large Language Models." ICLR 2021.
>
> [26] Sourab Mangrulkar, et al. PEFT: State-of-the-art Parameter-Efficient Fine-Tuning methods. 2022

---

> > ### Comment · Reviewer_1FBf · 2023-08-16
> >
> > Thank you very much for providing clarifications and additional supporting results.
> >
> > Would you please comment more on the infrastructure you used for training and training time?
> > Will you open source your code and data to ensure reproducibility?

---

> > > ### Author Response · Authors · 2023-08-16
> > >
> > > Thank you for the recognition of our rebuttal. For you questions:
> > >
> > > > Would you please comment more on the infrastructure you used for training and training time?
> > >
> > > Sure. We run our experiments on a HPC cluster, where each node has $8$ GPUs, $500$ GB CPU memory and $80$ cores. Our main experimental GPUs are Tesla V100s, $32$ GB GPU memory, as described in Sec 4 of our paper. Our software infrastructure is pytorch and fairseq. We use FP16 training.
> > >
> > > Each successful pretraining run (one hyper-parameter configuration) takes $24-32$ hours on $32$ V100s, ~ $1000$ GPU hours. The $32$ GPUs are spread on $4$ nodes with $8$ gpus on each node. Each language adaptation run takes the same time as one pretraining run except that we have to do it for all the languages. Each task adaptation run takes $6-12$ hours on $1$ V100  for each of the three tasks.
> > >
> > > > Will you open source your code and data to ensure reproducibility?
> > >
> > > Yes, we will open-source code. As for data, we use [CC100](https://data.statmt.org/cc-100/) for our pretraining and language adaptation (pretrained on English then adapted to a target language), which are publicly available. Our evaluation data are also public benchmarks: MLQA, XQuAD, XNLI. We will release relevant preprocessing scripts on these datasets for fostering reproducibility.

---

> > > > ### Comment · Reviewer_1FBf · 2023-08-17
> > > >
> > > > Thank you very much for the additional details.
> > > > The rebuttal addressed most of my concerns, hence I raised the score to reflect this.

---

### Official Review · Reviewer_Divy · 2023-07-06

**Soundness:** 3 good
**Presentation:** 3 good
**Contribution:** 3 good
**Rating:** 7
**Confidence:** 3

**Summary:**

This work introduces a training technique that leverages actively resetting token embedding to improve zero-shot language transfer. Experiments on RoBERTa show consistent improvement in multiple languages, distant languages in particular.

**Strengths:**

1. Simple and innovative approach.
2. Consistent improvement across languages and tasks.
3. Insightful analysis.

**Weaknesses:**

1. Only experimented on one pretrained model.


**Questions:**

How sensitive is this method to the choice of forgetting frequency?

---

> ### Author Rebuttal · Authors · 2023-08-09
>
> Thank you for your review.
>
> ### We appreciate that you recognize the simplicity and effectiveness of our method. We address your comments as follows.
>
> > Only experimented on one pretrained model.
>
> We would like to extend our experiments to more pretrained models. However, we are limited by the computational resources. One compiling of the entire experimental pipeline require at least 2 pretraining runs, 48 language adaptation runs, 6 task adaptation runs, which takes about 38460 GPU hours (V100 32GB).
>
> On the other hand, we chose RoBERTa as it is one of the most widely-used open-sourced pretrained language models. The effectiveness of our method on RoBERTa shows the potential of our method for the transformer family, as our method is not specific to a particular RoBERTa architecture but generalizable to any language model with a token embedding layer. We will open-source our code to facilitate future research on other pretrained models.
>
>
> Q:
> > How sensitive is this method to the choice of forgetting frequency?
>
> Please refer to the general response.

---

> > ### Comment · Reviewer_Divy · 2023-08-18
> >
> > Thanks for the explanation.

---

### Official Review · Reviewer_TTi1 · 2023-07-07

**Soundness:** 2 fair
**Presentation:** 2 fair
**Contribution:** 3 good
**Rating:** 6
**Confidence:** 4

**Summary:**

This paper follows the language adaptation procedure of MonoTrans (Artetxe et al., 2020), and proposes a new pre-training method with active forgetting. By resetting the embedding layer every K updates during training, the language model learns to learn the new embedding fast, similar to a meta-learning effect.

The paper conducted experiments on several languages, and evaluate the adapted models on zero-shot XNLI, MLQA and XQuAD. Unlike the experimental setting of MonoTrans, the paper considers a low-resource pre-training setting where an "unseen" language has as few as 5 million tokens for the adaptation step. The results show that the pre-trained models with active forgetting pre-training converge quickly during language adaptation, and outperform the baseline models.

**Strengths:**

The paper studies an important research question, i.e., language adaptation in a low-resource setting. Since the success of language models relies on large-scale pre-training, how to adapt the language models to low-resource languages without large-scale training data is beneficial for low-resource NLP research.

The proposed active forgetting is simple and effective for the following language adaptation step. Experiments demonstrate that it outperforms baseline models for zero-shot classification and question answering tasks.

**Weaknesses:**

- I am confused about goals and means:

  - The goal of language adaptation / MonoTrans (Artetxe et al., 2020) is to transfer existing high-resource language models (English in particular) to unseen languages, and the method is to learn language-specific word embeddings, i.e., language adaptation.

  - The goal of this paper seems to improve language adaptation results, rather than the goal behind language adaptation. The key is, transferring the currently existing models rather than pre-training new models. I understand the proposed method can benefit language adaptation. However, it greatly limits the scope, only the pre-trained models with the active forgetting can have this effect, i.e., currently existing pre-trained models are out of scope.

- Experimental setup. As mentioned in L273, the paper claims that multilingual pre-trained language models have issues that they need large corpora. Since many studies have shown that multilingual models have cross-lingual transferability, training multilingual language models in the low-resource setting should be a baseline.

- It would be great to provide the pre-training loss curves, showing how active forgetting affects the pre-training procedure.

- (Minor)  L192, Page7 footnote, missing ".". Figure 6, "XNLI Accuracy vs Adaptation Steps" is too small.

**Questions:**

- If active forgetting even requires a re-pretraining step, how to adapt currently existing pre-trained models without re-training them?

---

> ### Author Rebuttal · Authors · 2023-08-09
>
> Thank you for your constructive comments. We would like to address your concerns as follows.
> ### First, we would like to clarify our goal, motivation and contributions.
> > The goal of this paper seems to improve language adaptation results, rather than the goal behind language adaptation. The key is, transferring the currently existing models rather than pre-training new models.
>
> We agree with you that it is important to study various ways of transferring existing pretrained models. And there are already plenty of nice works in this line, e.g. adapters[2,13], regularization[11] and many others. We have discussed them in the related work section. However, the `transferability` of existing standard pretrained models is limited, causing non-trivial efforts to adapt/extend these models to new languages[2,11-13].
>
> An alternative way would be improving pretraining technology so that we can make more transferrable pretrained language models (PLMs). This line of research receives little attention despite its importance in alleviating the expensive cost of data and compute in downstream adaptation. As the field will see more pretrained models coming out in the following years, we argue that now is a good timing to consider such new pretraining technology. Because we are running out of high-quality data and language models are getting bigger and bigger.
>
> This is highly relevant to the plasticity research [3-10], where plasticity is defned as changing model prediction **with as little new information as possible**. In the context of language adaptation, pretrained models with "plasticity" should adapt to new languages with as little data in the new language as possible, and therefore reduce the cost of data in downstream adaptation.
>
> To sum up, our goal is not to reach another SoTA in language adaptation but rather using language adaptation as **a testing bed for understanding plasticity of language models**. We bring together efforts from different communities[3-10] and show that pretraining with active forgetting can be promising for improving language models' plasticity.
> ### Second, we want to elaborate on our experimental setup
> > Since many studies have shown that multilingual models have cross-lingual transferability, training multilingual language models in the low-resource setting should be a baseline.
>
> We agree that multilingual PLMs can be a meaningful baseline for grounding our numbers. We are training a multilingual pretrained baseline. We will update the results here once they are ready in the following days.
>
> Beyond that, we would like to emphasise that our work tackles a different scenario. We aim to have a flexible language model. No matter the pretraining corpus is monolingual or multilingual, this language model should easily generalise itself to  **unseen languages**. This is different from the scenario of multilingual PLMs like XLM-R[1], which requires seeing all the data for all languages from the scratch. Once done with pretraining and there is some new language distant from the pretraining languages you want to support, the multilingual PLMs might still struggle with **zero-shot transfer** as shown in several low-resources language research[15-17].
>
> ### Other Concerns and Questions
> > It would be great to provide the pre-training loss curves, showing how active forgetting affects the pre-training procedure.
>
> Please see the general response. The active forgetting creates an episodic learning pattern during pretraining, which is often seen in reinforcement learning or meta-learning.
>
> > (Minor) L192, Page7 footnote, missing ".". Figure 6, "XNLI Accuracy vs Adaptation Steps" is too small.
> Thank you, we will fix them in the camera-ready version.
>
> Q:
> >  If active forgetting even requires a re-pretraining step, how to adapt currently existing pre-trained models without re-training them?
> Thank you, this is indeed a valuable question.
>
> As discussed in our related work section and Part 1, there are many ways to adapt existing pretrained models to new languages according to prior research [2,11,13]. Our contribution is rather on the pretraining side. We hope our work can inspire research institutions/companies to **improve pretraining technology** and deliver PLMs with more "transferability". In this way the downsteam users can adapt them cheaply and without too many tweaking.
>
> On the other hand, it could be very exciting to take an existing model and make it perform active forgetting. We will leave this for future work since it is beyond the scope of this paper and requires further exploration.
>
> -----
>
>  References 1-14 can be found in our response to Reviewer ar7w.
>
> [15] Ebrahimi, Abteen, et al. "AmericasNLI: Evaluating Zero-shot Natural Language Understanding of Pretrained Multilingual Models in Truly Low-resource Languages." ACL 2022.
>
> [16] Adelani, David Ifeoluwa, et al. "MasakhaNER: Named Entity Recognition for African Languages." TACL 2021.
>
> [17] Adelani, David, et al. "MasakhaNER 2.0: Africa-centric Transfer Learning for Named Entity Recognition." EMNLP 2022.

---

> > ### Comment · Reviewer_TTi1 · 2023-08-20
> >
> > Thanks for addressing my concerns about the re-pretraining issue and the experimental setup. I have updated the score.

---

### Official Review · Reviewer_ar7w · 2023-07-09

**Soundness:** 3 good
**Presentation:** 3 good
**Contribution:** 2 fair
**Rating:** 4
**Confidence:** 4

**Summary:**

This paper presents an embedding forgetting mechanism for pre-training, aimed at enhancing robustness in downstream shift embedding fine-tuning. Focusing on the low-resource regime, the study conducts experiments on 10 simulated low-resource languages across three tasks: XNLI, XQUAD, and MLQA.

**Strengths:**

(1) This paper introduces a simple yet effective method, akin to a form of regularization, for improving the multilingual capabilities of a PLM. It demonstrates impressive performance when adapting to downstream X-tasks.
(2) The writing is clear, and the illustrative figures are informative.

**Weaknesses:**

(1) The paper is somewhat limited in scope, as it only applies to low-resource multilingual tasks within the framework of Artetxe et al. [2020].
(2) To propose a simple yet effective method, I believe solid experiments or rigorous proof are necessary; however, both are missing. In the experimental section, there is a lack of comprehensive ablation studies (e.g., why set low-resource data at 5M? Is this the real scenario in the multilingual setting? Why not use actual low-resource data instead of simulating one, considering that real low-resource data is different from sampled ones? How does the update frequency affect the results?).
(3) The paper lacks comparisons with other established methods, such as multilingual pre-training, multilingual adapters, and multilingual regularization techniques, among others.

**Questions:**

(1) Did you try different hyper-parameters for standard PLM and forgetting PLM when fine-tuning?
(2) Did you tie the weights of embedding and LM head?
Thanks for authors' detailed rebuttal, which addressed most of my concerns. However, the weaknesses are still existing. I'll keep my score.

**Limitations:**

The authors have discussed the limitations.

---

> ### Author Rebuttal · Authors · 2023-08-09
>
> Thanks for the review. We want to clarify a few misunderstandings and address your concerns as follows.
> ### First, we want to revisit and emphasise the scope of this work.
> > (1) The paper is somewhat limited in scope, as it only applies to low-resource multilingual tasks within the framework of Artetxe et al. [2020].
>
> Our motivation is to improve language models' plasticity. Plasticity of neural networks have been studied in graph learning, computer vision and reinforcement learning[3-10], where forgetting-relearn methods show promise. Our goal is to study plasticity in the context of pretrained language models. We believe this is a emerging research direction and will thrive in the following years.
>
> However, translating the `plasticity` concept to the language model setting is not trivial due to the lack of clear experimental setups. We note that, despite the model differences, almost all language models begins with a token embedding layer. As often tied to a specific vocabulary, the token embedding layer limits the plasticity, preventing generalisation to a new vocabulary.
>
> This observation inspires us to explore the plasticity of language models by manipulating the token embedding layer. Artetxe et al. [2020] draws our attention as it offers a nice experimental framework of **only manipulating the token embedding layer** for adapting between languages. We are not trying to improve SoTA multilingual models but rather testing if pretraining with active forgetting can be promising.
>
> In this sense, our work is rather well-scoped instead of "limited in scope", if contextualised in the line of plasticity research[3-10]. Our choice of Artetxe et al. [2020] as the experimental framework is also well-justified.  We will elaborate on this in our camera-ready version.
>
>
> ### Second, we want to address your concerns about our low-data experimental setup.
> > Why not use actual low-resource data instead of simulating one
> >
> > Why set low-resource data at 5M?
>
> We acknowledge that dealing with real-world low-resources languages can be more challenging than the low-data setup in our paper. We recognize there are rich work in this space: multilingual pretraining[1,12], multilingual adapters[2,13], and multilingual regularization[11]. We have discussed many of them in the related work section. However, the challenge of "low-resources" involve **multiple entangled factors**: the quality of the tokeniser, the amount of data, whether the script/language family is distant to the pretraining language(s) etc.
>
> **Simulating allows us to control these factors and isolate the effects of the factor that we are interested in** -- the amount of data in the new language. This factor is essential to our work as our goal is `plasticity` i.e. rewiring model prediction with as little new information as possible. Simulating various amount of data in the new language allows us to compare model plasticity as shown in Fig 3 in our paper.
>
> We observe forgetting PLMs outperform standard PLMs, when data is between 10K and 5M tokens. Since results on 5M is already decent enough to demonstrate the effectiveness of forgetting and most low-resources languages contain fewer than several million tokens, we chose to report the results on 5M. The curve in Fig 3 shows the results on other data amount.
>
> In summary, the choice of simulating a low-data well suits our research goal and allows us to contribute a clean piece of knowledge in the line of plasticity research.
>
> ### Third, we believe our experiments are comprehensive and convincing.
> > the study conducts experiments on 10 simulated low-resource languages
>
> We evaluated our method on three widely used cross-lingual transfer benchmarks XNLI, XQUAD, and MLQA, encompassing a variety of languages and two different types of tasks, NLI and QA. As shown in Tab 1 in our paper, XNLI contains 14 languages. Therefore we disagree with your comment that we only experiment with 10 languages. This is an unfair comment about our work.
>
> ### Other Concerns and Questions
> > How does the update frequency affect the results?
>
> We assume you mean the forgetting frequency $K$. If so, please see the general response.
>
> **Q1**: For fair comparison, we use the same hyper-parameters following previous work Kelly 2022 and RoBERTa. We will open-source the code.
>
> **Q2**: The embeddings and the weights in LM head are tied as described in RoBERTa paper[14]
>
> ------
>
> [1] Conneau, Alexis, et al. "Unsupervised Cross-lingual Representation Learning at Scale. ACL 2020.
>
> [2] Pfeiffer, Jonas, et al. "MAD-X: An Adapter-Based Framework for Multi-Task Cross-Lingual Transfer." EMNLP 2020.
>
> [3] Lyle, Clare, et al. "Understanding Plasticity in Neural Networks." ICML 2023
>
> [4] Zhou, Hattie, et al. Fortuitous forgetting in connectionist networks. ICLR 2022
>
> [5] Chen, Yihong, et al. Refactor gnns: Revisiting factorisation-based models from a message-passing perspective. NeurIPS 2022.
>
> [6] Igl, Maximilian, et al. "Transient Non-stationarity and Generalisation in Deep Reinforcement Learning." ICLR 2020.
>
> [7] Alabdulmohsin,  Ibrahim, et al. The impact of reinitialization on generalization in convolutional neural networks. arXiv:2109.00267, 2021.
>
> [8] Taha, Ahmed, et al. Knowledge evolution in neural networks. CVPR 2021.
>
> [9] D’Oro, Pierluca, et al.  Sample-efficient reinforcement learning by breaking the replay ratio barrier. In Deep Reinforcement Learning Workshop NeurIPS 2022.
>
> [10] Nikishin, Evgenii, et al. The primacy bias in deep reinforcement learning. ICML 2022.
>
> [11] Pfeiffer, Jonas, et al. Unks everywhere: Adapting multilingual language models to new scripts. EMNLP 2021.
>
> [12] Pfeiffer, Jonas, et al. Lifting the curse of multilinguality by pre-training modular transformers. NAACL 2022.
>
> [13] Ansell, Alan, et al. Composable sparse fine-tuning for cross-lingual transfer. ACL 2022.
>
> [14] Liu, Yinhan, et al. Roberta: A robustly optimized bert pretraining approach." arXiv:1907.11692 (2019).

---

### Author Rebuttal · Authors · 2023-08-10

We thank all the reviewers for their valuable comments. We would like to address some common questions in this general response. **Figures are attached in the rebuttal pdf.**

### Active Forgetting Creates An Episodic Learning Pattern
Reviewers are curious about the loss curves comparison for standard pretraining and forgetting pretraining. Hence we have included the loss curves for standard pretraining and forgetting pretraining in Rebuttal Figure 1. We can see that the forgetting pretraining creates a very interesting pattern of loss curves. If we zoom in a bit, as shown in Rebuttal Figure 2, episodic learning is happening while we are actually using the same data for pretraining. This is quite different from introducing diversity by including as many languages as possible in the pretraining corpus. It is more like in reinforcement learning or meta-learning.

### Impact of Forgetting Frequency
We would like to elaborate on our choice of forgetting frequency $K$. In our preliminary experiments, we tried $K=100, 1000, 5000$. We find $K=1000$ works well and thus sticks with it. Since we don't want to overtune the hyperparameters, we just use the same $K$ for all the experiments.
We include the loss curves of $K=100$ and $K=5000$ here. We can see that both forgetting too frequently and forgetting too infrequently will hurt the performance. Too frequent forgetting leaves little time for the body to learn something meaningful (the pretraining loss stuck around 11). Too sparse forgetting will make the body hard to adjust to the next forgetting, causing divergence as pretraining goes on.

---

> ### Author Response · Authors · 2023-08-10
>
> ### Multilingual Pretraining
> We ran additional experiments with multilingual pretraining as suggested by the reviewers. Here is how we did it
> - For a fair comparion, we trained a multilingual RoBERTa-base with the same model size as our monolingual model. We also subsample the multilingual corpus, CC100, to ensure that the pretraining corpus is of the same size as our monolingual pretraining corpus.
>     - Language Emb/Task Body Adaptation refers to separately adapting embeddings with 5M tokens of _Thai_ unlabelled data and adapting body with English NLI data.
>     - Task Full Model Adaptation refers to adapting the full model with English NLI data
>     - Note that **_Thai_ is already included in multilingual CC100 (6B tokens in the original dataset, 720M tokens in our subsampled dataset)**
>
> We can see that multilingual pretraining indeed helps cross-lingual transfer *when the language is in the pretraining data*. On the other hand, we can also observe that forgetting indeed lifts the adaptation performance:
> - Comparing Row 2 and Row 5 (59.7 vs 60.0), we can see that even with as little as 5M unsupervised data,  **the Forget model is almost as competitive as its multilingual counterpart, despite only pretrained on monolingual data.**
> - Comparing Row 3 and Row 4 (49.4 vs 55.0), we can see that, **forgetting also helps adapt multilingual pretrained models**.
> - Comparing Row 1 and Row 2 (35.3 vs 59.7), we can see that **forgetting helps monolingual pretrained models a lot**
>
> XLM-R (base) outperform best our multilingual pretrained baselines (72.4 vs 60.0). This is no surprise due to its large pretraining corpus (10x our multilingual corpora) and model size (2x our multilingual model).
>
> We hope these results can help reviewers ground our numbers better.
>
> -----
>
> | Pretrain Corpus          | #Languages In Pretraining Corpus | #parameters | Pretraining Method      | Adaptation Framework              | Zero-shot Thai XNLI Acc |
> |--------------------------|----------------------------------|-------------|-------------------------|-----------------------------------|-------------------------|
> | 300GB English CC100      | 1                                | 125M        | Standard-RoBERTa (base) | Language Emb/Task Body Adaptation | 35.3                    |
> | 300GB English CC100      | 1                                | 125M        | Forget-RoBERTa (base)   | Language Emb/Task Body Adaptation | 59.7                    |
> | 300GB Multilingual CC100 | 100                              | 125M        | Standard-RoBERTa (base) | Language Emb/Task Body Adaptation | 49.4                    |
> | 300GB Multilingual CC100 | 100                              | 125M        | Forget-RoBERTa (base)   | Language Emb/Task Body Adaptation | 55.0                    |
> | 300GB Multilingual CC100 | 100                              | 125M        | Standard-RoBERTa (base) | Task Full Model Adaptation        | 60.0                    |
> | 2.5TB Multilingual CC100 | 100                              | 270M        | XLM-RoBERTa (base)      | Task Full Model Adaptation        | 72.4                    |

---

### Decision · Program_Chairs · 2023-09-21

**Decision:**

Accept (poster)

**Comment:**

We note that reviewer ar7w has not acknowledged the author response. However, the author response has largely addressed the reviewers’ concerns. As the authors explain, this paper is not aiming at "improving SoTA multilingual models but rather testing if pretraining with active forgetting can be promising". Indeed this seems like a promising approach. The authors present a novel technical contribution to the community that is experimentally demonstrated to be effective and improve LM plasticity. Two key issues that are raised are the lack of ablation experiments (Reviewer 1FBf) and the need for additional baselines, both of which would give crucial and further insight into the effectiveness of the approach. The authors address these by running / presenting new experiments albeit only limited ones (ie, on Arabic and Thai respectively). The paper would benefit further from more comprehensive experiments in that respect, and, in particular, showing not only detailed results per language, but also aggregated (Avg) results (some of which could go in the Appendix given the space restrictions). The authors are asked to provide these experiments in the camera ready.